# Recognizing Urban Functional Zones by GF-7 Satellite Stereo Imagery and POI Data

Zhenhui Sun [1,2,†], Peihang Li [1,2,*,†], Dongchuan Wang [1,2], Qingyan Meng [3,4,5], Yunxiao Sun [1] and Weifeng Zhai [6]

1   School of Geology and Geomatics, Tianjin Chengjian University, Tianjin 300384, China; sunzhcas@gmail.com (Z.S.); mrwangdc@126.com (D.W.); sunyx@tcu.edu.cn (Y.S.)
2   Tianjin Key Laboratory of Soft Soil Characteristics and Engineering Environment, Tianjin University, Tianjin 300384, China
3   Aerospace Information Research Institute, Chinese Academy of Sciences, Beijing 100101, China; mengqy@radi.ac.cn
4   University of Chinese Academy of Sciences, Beijing 100049, China
5   Key Laboratory of Earth Observation of Hainan Province, Hainan Aerospace Information Research Institute, Sanya 572029, China
6   School of Sciences, Qiqihar University, Qiqihar 161006, China; weifengzhai@sohu.com
*   Correspondence: lph5878@163.com
†   These authors contributed equally to this work.

**Abstract:** The identification of urban functional zones (UFZs) is crucial for urban planning and optimizing industrial layout. Fusing remote sensing images and social perception data is an effective way to identify UFZs. Previous studies on UFZs recognition often ignored band information outside the red–green–blue (RGB), especially three-dimensional (3D) urban morphology information. In addition, the probabilistic methods ignore the potential semantic information of Point of Interest (POI) data. Therefore, we propose an "Image + Text" multimodal data fusion framework for UFZs recognition. To effectively utilize the information of Gaofen-7(GF-7) stereo images, we designed a semi-transfer UFZs recognition model. The transferred model uses the pre-trained model to extract the deep features from RGB images, and a small self-built convolutional network is designed to extract the features from RGB bands, near-infrared (NIR) band, and normalized digital surface model (nDSM) generated by GF-7. Latent Dirichlet allocation (LDA) is employed to extract POI semantic features. The fusion features of the deep features of the GF-7 image and the semantic features of POI are fed into a classifier to identify UFZs. The experimental results show that: (1) The highest overall accuracy of 88.17% and the highest kappa coefficient of 83.91% are obtained in the Beijing Fourth Ring District. (2) nDSM and NIR data improve the overall accuracy of UFZs identification. (3) POI data significantly enhance the recognition accuracy of UFZs, except for shantytowns. This UFZs identification is simple and easy to implement, which can provide a reference for related research. However, considering the availability of POI data distribution, other data with socioeconomic attributes should be considered, and other multimodal fusion strategies are worth exploring in the future.

**Keywords:** data fusion; GF-7 image; POI; 3D urban morphology; urban functional zones

## 1. Introduction

As the basic units of urban, UFZs are the spatial carriers of various social and economic activities, such as commercial zones, residential zones, and industrial zones [1–3]. With the rapid expansion of cities, a large number of people and means of production gather in the cities, resulting in an increasingly complex urban functional structure. Therefore, the accurate and rapid identification of UFZs holds substantial research and application value. It aids in understanding the interplay between human spatial activities and socioeconomic operations [4]. Moreover, the spatial distribution of urban functions can be utilized to guide

urban planning management, resource allocation, environmental monitoring, population estimation, and other related activities [5–7].

Traditional methods for identifying UFZs have relied on statistical investigation and expert judgment. However, these methods often require field investigation and human judgment, which are highly subjective and time-consuming [8,9]. In recent decades, with the rapid development of remote sensing technology, remote sensing images have been widely used in urban research fields, including land cover mapping, environmental protection, and deformation monitoring [10–12]. These studies provide new ideas for the identification of UFZs. Based on different image feature extraction methods, UFZs recognition methods can be broadly divided into two categories. The first category is based on artificially designed features. Some methods employ probabilistic topic mode (PTM) to classify UFZs or scenes by extracting shallow features from images, such as spectral, texture, structure, scale-invariant feature transform (SIFT), and other features [13–17]. Other methods involve constructing urban landscape features by obtaining urban land cover maps through remote sensing images and then using machine learning techniques to identify UFZs [18–20]. However, this type of method requires researchers to possess extensive experience and knowledge, to determine the best feature combination through multiple experiments. Moreover, the optimal feature combination often lacks effective transferability between different cities, limiting its generalization. The second category focuses on automatically extracting image features using deep learning methods. With the wide application of deep learning technology in the field of remote sensing, its advantages in identifying UFZs have gradually become prominent [21–25]. However, most pre-trained models only accept RGB images, while high-resolution remote sensing satellites can acquire RGB + NIR multi-spectral images, and even some satellites provide multi-view and more band information, which limits the direct use of the "pre-training and fine-tuning" method for model training on multi-spectral images. To address this challenge, Huang et al. [26] designed a semi-transferred convolutional neural network for UFZs recognition. However, their method relied on the AlexNet pre-training model [27], without considering the impact of other pre-training models on UFZ identification. Additionally, their classification of UFZs was based solely on remote sensing images, so it needs further improvement.

Remote sensing images provide information about the natural landscape of cities, but they lack detailed insight into human economic activities in detail. To overcome this limitation, previous studies have used social perception data such as social media data, traffic data, Weibo check-in data, and POI data to identify UFZs. This type of data can effectively reflect the socioeconomic attributes of urban functional units and finely identify different types of UFZs [28–32]. Among these data sources, POI data are particularly useful as they can be freely obtained from Internet maps without infringing on user privacy. It has been widely used in the identification of UFZs. However, it is important to note that POI data have limitations due to the uneven distribution and biased categorization of human activities [1]. Fusing remote sensing images and social perception data is an effective way to quickly and accurately identify UFZs.

From the perspective of data, most studies on identifying UFZs focus on two-dimensional (2D) data, overlooking the importance of 3D urban morphology information. However, certain urban functional units have differences in urban morphology information, such as office business districts, shopping malls, residential quarters, and shantytowns. Several previous studies have demonstrated that incorporating 3D urban morphology can enhance the accuracy of UFZs identification. The generation of 3D urban information typically relies on building vector data obtained from Internet maps or government departments. However, due to differences in data collection time or quality, these vector data may not be entirely consistent with the actual situation and ignore the 3D information of other ground objects except buildings [33–35]. Some researchers, such as Zhao et al. [36], have utilized the building height information estimated from building shadows to classify urban building functions, improving the recognition accuracy of residential buildings. Nonetheless, this method depends on parameters, such as the sun and satellite altitude, and is not universally

applicable due to irregularities in shadow shape area, adhesion, and other phenomena. Airborne Lidar data have been employed to construct 3D urban structure parameters for UFZs classification and achieved good results [37,38]. However, the collection of airborne Lidar data collection is costly and its coverage is limited. Huang et al. [39] introduced ZY-3 multi-view images into the model to describe the 3D urban morphology information, demonstrating the helpfulness of multi-view images in UFZs identification. Nevertheless, this method does not directly account for the influence of ground objects' height on the UFZs identification. Currently, several high-resolution stereoscopic mapping satellites have been launched, enabling the rapid acquisition of digital surface model (DSM) [40–42]. GF-7 is one of the most advanced stereo observation satellites in China and generates DSM data with an elevation Root Mean Square Error (RMSE) within 1m, providing large-scale and detailed three-dimensional information about urban ground objects [43]. The fusion of GF-7 multi-spectral images and panchromatic stereoscopic images effectively captures the 2D and 3D features of urban objects, harnessing the advantages of GF-7 multi-view images [44,45].

In summary, we proposed a simple and effective "Image + Text" fusion strategy for UFZs recognition, utilizing GF-7 multi-view and multi-spectral images and POI data. The main contributions are as follows:

(1) We extract image features using a semi-transfer learning strategy and employ the LDA topic model to generate POI semantic features. These features are then fused to improve UFZ recognition.
(2) We incorporate both the 2D and 3D characteristics of the study area.
(3) We investigate the function of DSM generated from GF-7 images, NIR, and POI in identifying UFZs.

## 2. Study Area and Data

### 2.1. Study Area

Beijing is the capital of China and serves as the political, economic, and cultural center, with a high urbanization rate of approximately 86.5%. For our study, we specifically selected the area within the Fourth Ring Road of Beijing, as shown in Figure 1, which spans an approximate area of 300 km$^2$. The study area encompasses significant commercial zones, residential zones, and numerous administrative agencies in Beijing. The diverse and mixed functions within this area pose challenges for accurately identifying UFZs.

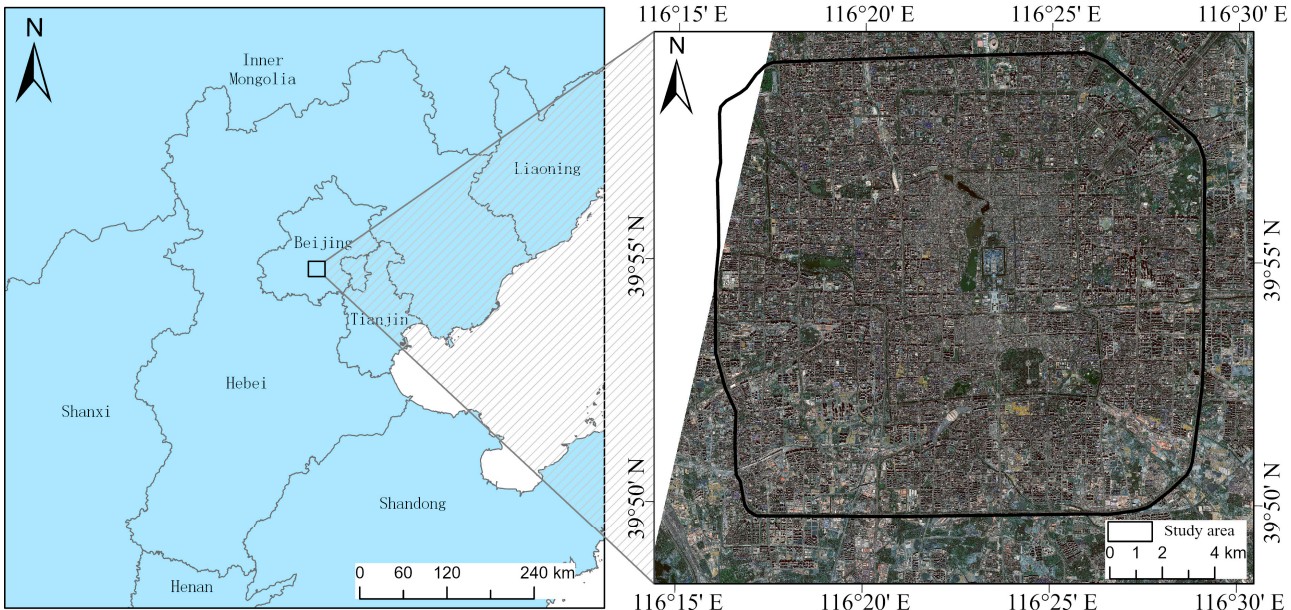

**Figure 1.** Location of the study area.

### 2.2. Data

The GF-7 is China's first civil optical stereo mapping satellite with a sub-meter resolution. It was successfully launched in November 2019 with an orbital altitude of about 506 km and has a return cycle period of fewer than 60 days. It can effectively acquire 20 km wide front-view panchromatic stereoscopic images with a 0.8 m resolution, rear-view panchromatic stereoscopic images with a 0.65 m resolution, and multi-spectral images with a 2.6 m resolution. The specific parameters of the GF-7 dual-line array stereo camera are shown in Table 1.

**Table 1.** Parameters of bi-linear array stereo camera.

| Spectral Band | Wavelength (μm) | Spatial Resolution (m) | Swath Width (km) |
|---|---|---|---|
| Front-view Pan | 0.45–0.90 | 0.8 | |
| Rear-view Pan | 0.45–0.90 | 0.65 | |
| Blue | 0.45–0.52 | | |
| Green | 0.52–0.59 | | ≥20 |
| Red | 0.63–0.69 | 2.6 | |
| NIR | 0.77–0.89 | | |

In this study, two adjacent GF-7 images were selected on 16 October 2020 with less clouds, a clear texture, and geometric structure, which effectively covered most of the area within the Beijing Fourth Ring Road. The 0.65 m multi-spectral image underwent preprocessing steps, such as orthorectification, image sharpening, and mosaicking, using PCI Geomatica 2021 SP4. Subsequently, the multi-spectral images were resampled to a resolution of 1.3 m as required. DSM data are generated by utilizing both the GF-7 front-view panchromatic image and rear-view panchromatic image. Initially, the rational function model was created, and then the ground control point (GCP) and tie point (TP) were collected to improve the mathematical model and ensure that the stereo pairs were aligned with each other. Finally, the semi-global matching (SGM) algorithm was used to derive 1 m DSM data [46]. Meanwhile, we take blocks and some buildings as spatial units, and a series of filtering algorithms available in PCI Geomatica were carefully filtered the DSM to obtain Digital Elevation Model (DEM). Then, DSM minus DEM to obtain nDSM, which is used to describe the relative height of ground objects.

Furthermore, due to water surface clutter and the building being occluded, the nDSM value of the water surface is wrong and some building height information is missing. We utilized the surrounding ground object height and building floor data as references to correct the nDSM data to obtain the finalized nDSM data. Figure 2 shows the finalized nDSM data.

POI data are a kind of spatial point data, which are an abstract expression of geographic entities in the real world. Any geographic entity can be represented by POIs, such as shopping malls, universities, residences, and parks, and typically include attributes, such as names, categories, and geographic coordinates. In this study, the POI data were obtained using the Gaode map API, and the data were downloaded on 10 July 2021. The Gaode map has three levels of classification for POIs, including 23 first-level classifications, 267 s-level classifications, and 869 third-level classifications. Some POI types in the first-level classifications contain different functional types, and POI types in the third-level classifications are too elaborate and not conducive to data processing. Therefore, we opted to utilize the second-level classification as our standard for classification and removed some POI categories that are easy to interfere with the identification of UFZs, such as shared equipment, ATMs, parking lots, and ticket offices. Consequently, a total of 381,905 POIs were obtained. The acquired POI data were transformed into the WGS84 coordinate system. The spatial distribution of the POI data is shown in Figure 3.

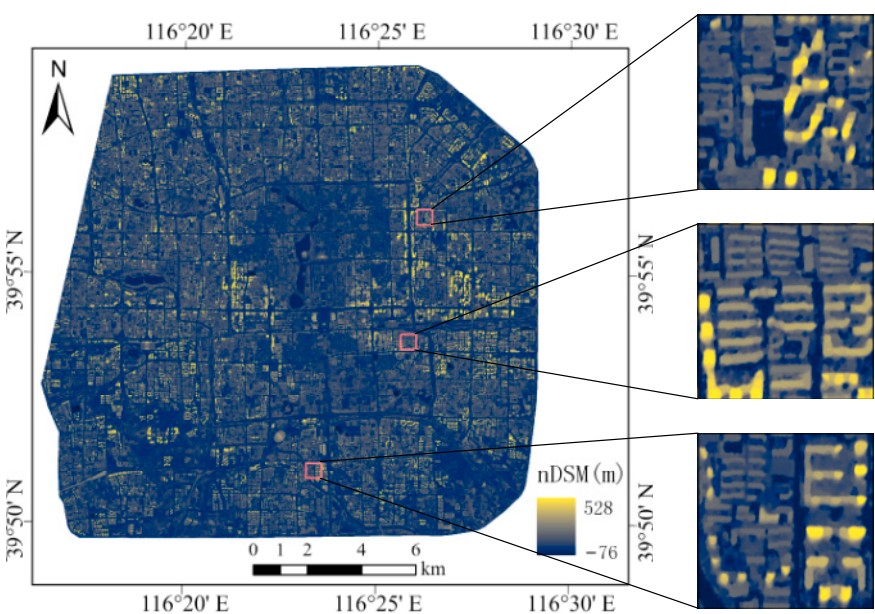

**Figure 2.** nDSM data of the study area.

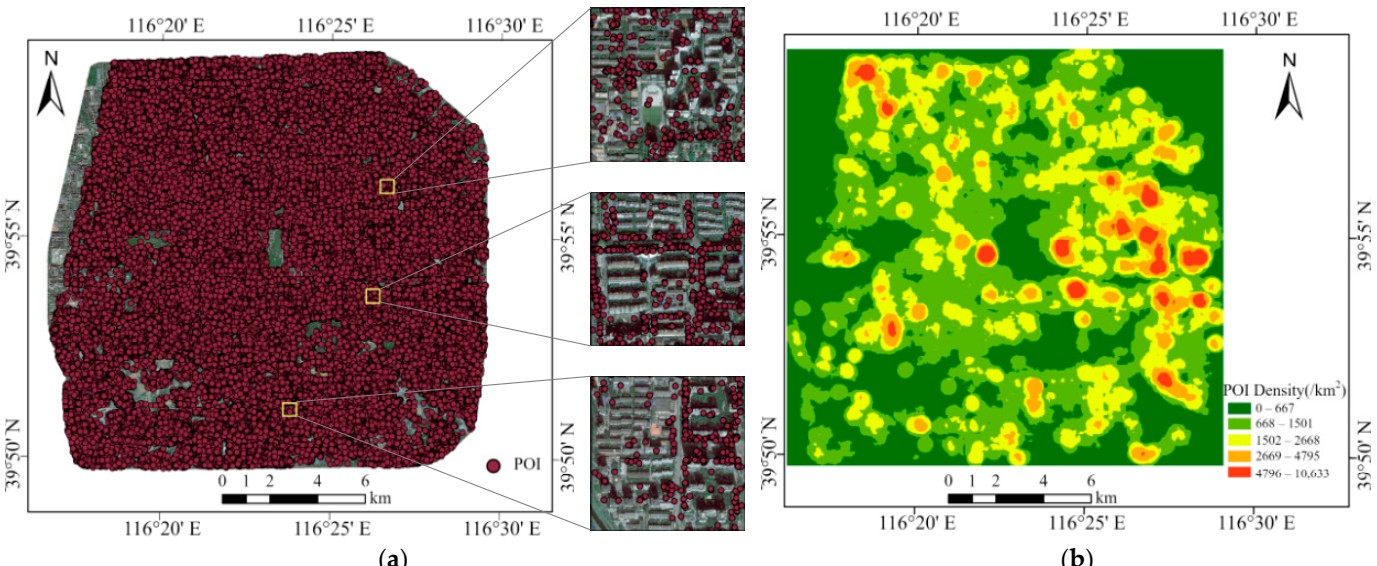

(a)　　　　　　　　　　　　　　　　　　　　　　　　　　(b)

**Figure 3.** POI data in the study area. (**a**) Spatial distribution of POI data. (**b**) Kernel density map of POI data.

OpenStreetMap (OSM) is a collaborative project that creates a free editable map database of the world, including vector data such as roads, water systems, lakes, green spaces, and building outlines, which has important reference value for the boundary division and type interpretation of urban functional units. The OSM road network data are used to divide urban functional units, downloaded on 24 March 2021.

### 2.3. UFZs Categories

Referring to relevant Chinese national standards (Code for Classification of Urban Land-Use and Planning Standards of Development Land GB 50137–2011, China), and considering the main land use types within the Fourth Ring Road of Beijing, seven types of UFZs were determined. The detailed categories schemes and descriptions of UFZs are provided in Table 2.

**Table 2.** Descriptions of UFZs categories schemes.

| Category | Descriptions |
|---|---|
| Residential zones | Regular, well-equipped communities, such as apartments and high-rise residential areas |
| Commercial zones | Commercial retail, restaurants, financial, and media places, such as office buildings and malls |
| Shantytown | Dilapidated, old low-rise communities, such as villages within cities |
| Public service | Administrative, medical, sport, and cultural places, such as government, hospitals, and libraries |
| Development | A place to be developed or under construction |
| Education | Education and research places, such as schools, universities, and institutes |
| Green land | Park and greenspace places, such as parks, greenbelts, and water |

## 3. Methodology

In this study, a multimodal data semi-transfer fusion framework called "Image + Text" is proposed for identifying UFZs by fusing GF-7 imagery and POI data, as shown in Figure 4. The framework includes the following three steps: (1) data preprocessing, including the division of urban functional units, the construction of remote sensing image datasets, and POI text datasets; (2) multimodal data feature extraction and fusion, including extraction and fusion of remote sensing image features and POI semantic features; (3) identification and mapping of UFZs, including preferred classifiers to identify UFZs and accuracy evaluation.

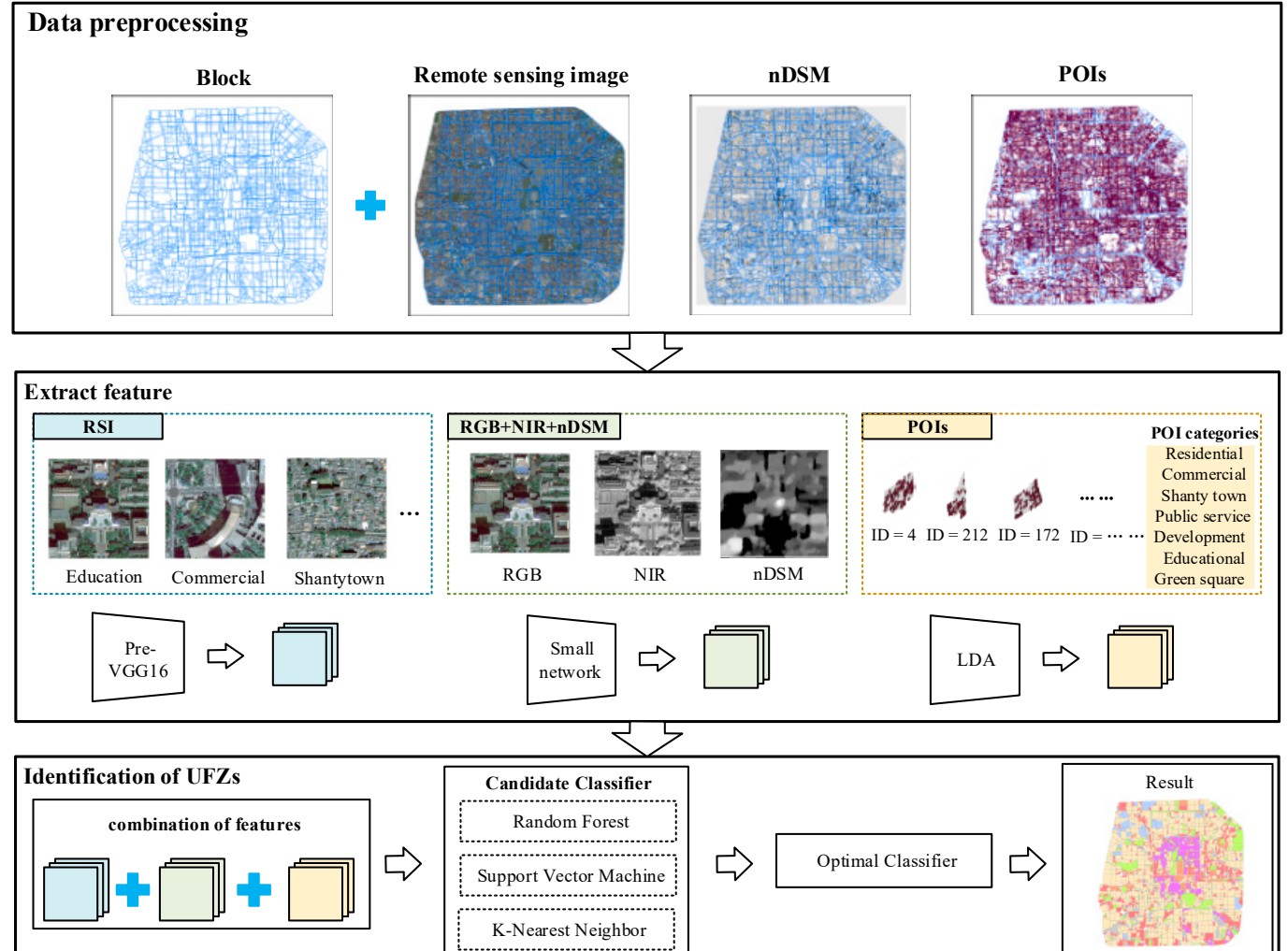

**Figure 4.** The process flow of UFZs identification.

### 3.1. Urban Functional Unit Division

The urban functional unit division methods include grid method, block method, traffic analysis zone method, and geoscene segmentation method [47–50]. The block method is simple and can describe the real boundaries of UFZs and is widely used in UFZs identification [51]. In this study, we utilize the blocks defined by the OSM road network as the basic unit for UFZs identification. To ensure precise block boundaries, we preprocess the OSM road network. First, low-level roads are removed and retain tertiary, secondary, primary, motorway, and trunk roads. We determine road widths for different levels and create road buffer zones using Chinese urban road standards and remote sensing images. The ArcGIS software's Erase function is then used to remove the road buffer zone polygons within the Fourth Ring Road to obtain separate urban functional units. We obtain a total of 1971 urban functional units, each assigned a unique ID. To refine the boundaries further, we consult Baidu maps to determine the functional types of the urban functional units and make necessary corrections. Figure 5 shows the process of urban functional unit division.

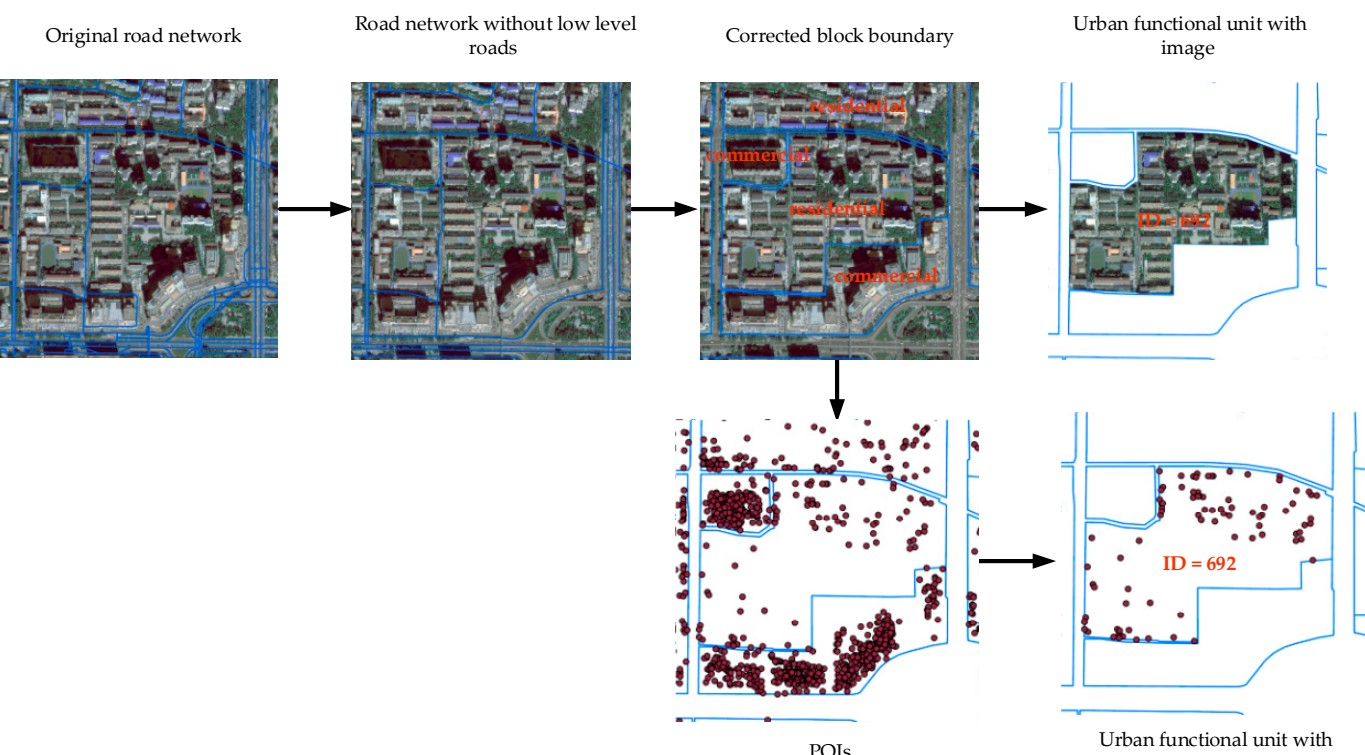

**Figure 5.** Urban functional unit division.

### 3.2. Image Dataset Generation

Remote sensing images divided by urban functional units are usually irregular and of varying sizes, which are not suitable as input for deep learning models. Therefore, we have designed a central window cropping strategy, selecting the center of the urban functional unit as the window center and cropping the 1.3 m image with a size of 227 × 227 pixels. These cropped images are used to generate the dataset. To address the imbalance in the number of samples for different UFZs, we employ strategies of diagonal offset cropping and image flipping for data augmentation on UFZs with a small number of samples. As shown in Figure 6, the red-dotted box represents the new window obtained by diagonally moving the central window toward the upper left corner and moving 20 pixels in both horizontal and vertical directions. The yellow-, green-, and blue-dotted boxes indicate the new windows resulting from translating the central window in the other three diagonal directions. The detailed composition of the dataset is shown in Table 3, the training data have 4240 images and the test data have 1006 images. Through image augmentation, the

number of training samples increases by more than 4 times, and the number of samples in each category is balanced effectively.

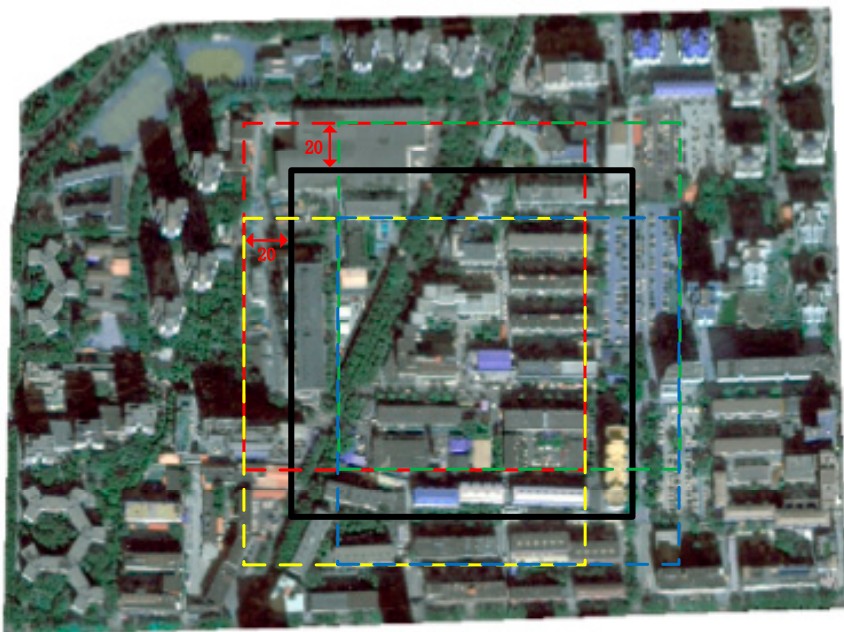

**Figure 6.** Image cropping using a central windowing strategy.

**Table 3.** UFZs self-built image dataset.

|  | Residence | Commercial | Shantytown | Public Service | Development | Education | Green Land | Total |
|---|---|---|---|---|---|---|---|---|
| Training set | 360 | 230 | 90 | 60 | 80 | 100 | 45 | 965 |
| Training set [1] | 720 | 690 | 720 | 600 | 560 | 500 | 450 | 4240 |
| Test set | 417 | 239 | 102 | 56 | 74 | 101 | 17 | 1006 |

[1] Augmented training set.

### 3.3. Multimodal Data Feature Extraction and Fusion

The proposed multimodal data fusion framework is shown in Figure 7, including image feature extraction, POI semantic feature extraction, and feature fusion. Image features are extracted from remote sensing images using a semi-transfer convolutional neural network (ST-CNN), semantic features are extracted from the POI data using the LDA topic model, and two parts of features are fused together.

#### 3.3.1. Image Feature Extraction

For the pre-trained model, the related parameters are initialized by the pre-trained VGG16. As the pre-trained VGG16 is trained by the images of size 224 × 224 pixels with RGB bands, the images of size 227 × 227 pixels with RGB bands are resized to 224 × 224 pixels and input the transfer DCNN. In addition, in order to match the number of POI semantic features and reduce the parameter of the model, we delete the network structure beyond the first fully connected layer and add a fully connected layer that outputs 64 dimensions.

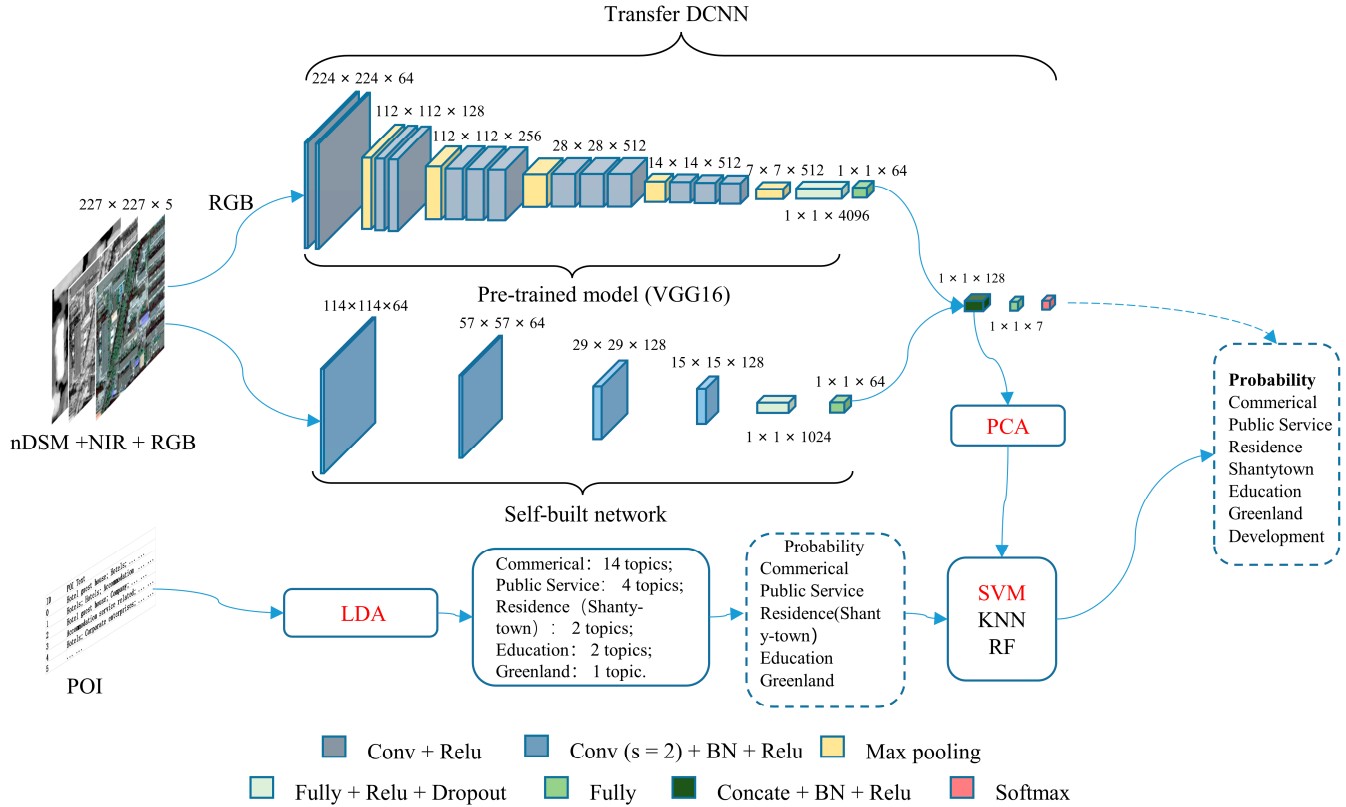

**Figure 7.** Multimodal data fusion model.

For the self-built network, four 3 × 3 convolutional layers with a stride of 2 are used, followed by two fully connected layers. The output dimensions of the first and second fully connected layers are set to 1024 and 64, respectively. The self-built network input RGB + NIR + nDSM images with 227 × 227 pixel size, to avoid the limitations of the pre-trained VGG16.

Finally, the outputs of the pre-trained model and the self-built CNN are fused into a single layer. To avoid the instability of the model caused by feature fusion, a Batch Normalization (BN) layer and a Rectified Linear Unit (Relu) layer are added after the fusion layer. A fully connected layer with an output dimension of 7 is designed to match the number of categories of UFZs. After the softmax function, the categories of UFZs are provided. UFZs samples are fed into the ST-CNN model to train it, resulting in model parameters specifically optimized for UFZs recognition being obtained. The feature fused by pre-trained VGG16 and the self-built CNN are regarded as image features, with an output dimension of 128.

### 3.3.2. Semantic Feature Extraction

POI data provide valuable insights into human economic activities. However, some previous studies have primarily focused on counting POI frequencies for text modeling, ignoring the potential semantic features [52–54]. LDA [55] is a three-layer Bayesian probability model, which can identify the semantic topic information in large-scale document sets or corpus. It has demonstrated promising results in identifying UFZs [56,57]. The number of topics is crucial for LDA. An appropriate number of topics can effectively prevent overfitting and obtain higher accurate results. We use the confusion algorithm to calculate the number of topics, and the smaller the perplexity, the more appropriate the determined number of topics is for LDA [55]. We utilize the secondary classification of Gaode POI to divide the functional categories of each POI data, the POI functional attributes within the same functional unit are regarded as a document in the LDA model. Figure 8 shows the perplexity values for 1 to 99 topics. When the topic number is 25, the perplexity is the

smallest. Combined with the UFZs category, the number of topics is finally determined to be 22 for LDA modeling; 14 topics correspond to a commercial zone, 4 topics correspond to public service, 2 topics correspond to a residential zone, 1 topic corresponds to education, and 1 topic corresponds to green land. Due to the proportion of corresponding topics in different urban functional units is not balanced, the semantic shrinkage method proposed by Xing et al. [58] is adopted to process the probability values of different types of topics in the same urban functional unit. Finally, each urban functional unit corresponds to the probability of five kinds of urban functions, both residential zone and shantytowns undertake residential functions and are combined into one topic. There are almost no POIs in development, and the topic of development is discarded.

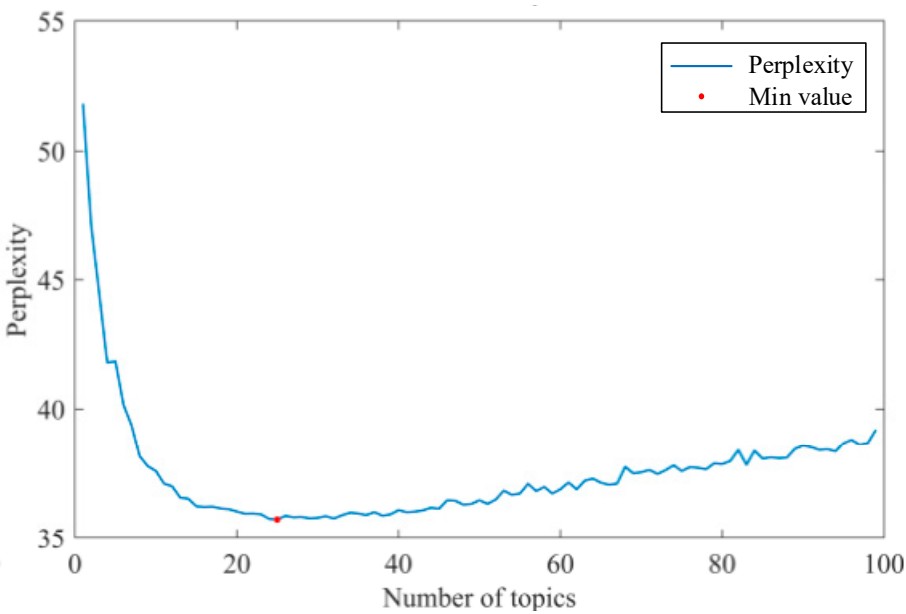

**Figure 8.** Perplexity values for the different number of topics.

### 3.3.3. Feature Fusion and UFZs Identification

The image features extracted by ST-CNN and the semantic features extracted by LDA are concatenated. The image feature dimension (128 dimensions) is about 25 times that of the POI semantic feature dimension (5 dimensions), and direct concatenating leads to fusion features that overly emphasize image features, it is necessary to reduce the dimensionality of image features. PCA is a dimension-reduction technique that linearly transforms the original space into a new space of smaller dimensions while simultaneously describing the variability of the data as much as possible [59]. PCA projects along the eigenvectors of the covariance matrix corresponding to the largest eigenvalues, where the eigenvectors point in the direction with the highest amount of data variation. In general, the first few principal components whose cumulative variance contribution exceeds 95% are considered dimensionality-reduced data and often contain nearly all information from the original data. In this study, PCA is employed to reduce the dimensionality of image features. The specific image dimension after dimensionality reduction is determined through experimentation. After reducing the dimensionality of the image features, they are concatenated with the semantic features. Following the feature fusion, three classifiers, namely, Support Vector Machine (SVM), K-Nearest Neighbors (KNN), and Random Forest (RF), are utilized for identifying UFZs [60–62].

### 3.4. Accuracy Evaluation

To quantitatively evaluate the classification performance of different models, the overall accuracy (OA) and kappa coefficient are utilized as model evaluation indicators. OA is defined as the ratio of the number of correctly classified data to the total test data. It provides an intuitive measure to assess the overall classification performance of the test data. The kappa coefficient is considered to be a more robust measure compared to a simple percent agreement calculation because it takes into account the possibility of the agreement occurring by chance.

## 4. Results and Discussion

### 4.1. Results

#### 4.1.1. Experimental Setup

The experiments were performed on a Windows Operating System, using CPU (Intel Core (TM) i5-11400F@2.60GHz), RAM (16 GB), and GPU (NVIDIA GeForce RTX 3060 12 GB). Experimental work for image feature extraction is performed using the Deep Learning Toolbox in MATLAB 2022b. The training data are randomly divided into training set and verification set according to the ratio of 4:1, and the test sets are used to verify the performance of the model. The learning rate of the pre-trained VGG16 in ST-CNN is $3 \times 10^{-5}$, and the learning rate of other layers is $3 \times 10^{-4}$. The adaptive moment estimation (Adam) algorithm is engaged in optimizing the model. The batch size is set as 32. The network with the pre-training model is trained for 20 batches. The network without the pre-training model is trained for 80 batches.

POI semantic information extraction, PCA dimension reduction, and RF, KNN, and SVM classifiers are all based on the Scikit-Learn library, and the programming language uses Python 3.7.

#### 4.1.2. UFZs Identification Results

The identification results of UFZs are shown in Figure 9, and the quantitative evaluation results are summarized in Table 4. Qualitative results show no significant difference in the performance of the different classifiers. Quantitative indicators OA and kappa coefficient are not much different. Among them, SVM outperforms the others with the highest OA and kappa coefficient, has obvious advantages in identifying public services and shantytowns. KNN follows with the second-best performance, particularly excelling in identifying green land and education. RF obtains the lowest OA and kappa coefficient, but it has obvious advantages in identifying commercial zones and development. For different categories of UFZs, the accuracy of identifying public services and education is relatively low, and the identification accuracy is mostly less than 70%. The accuracy of identifying residential zones, green land, and development surpasses 90%.

**Table 4.** Overall classification results and per category results. The bold values represent the best performance of each indicator.

|  | SVM (%) | KNN (%) | RF (%) |
| --- | --- | --- | --- |
| Residential zone | 94.00 | **94.72** | 91.61 |
| Commercial zone | 83.68 | 85.36 | **87.45** |
| Shantytown | **92.16** | 87.25 | 90.20 |
| Public service | **55.41** | 47.30 | 47.30 |
| Development | 96.43 | 94.64 | **98.21** |
| Education | 64.71 | **70.59** | 47.06 |
| Green land | 94.06 | **96.04** | 95.05 |
| OA | **88.17** | 87.97 | 87.18 |
| Kappa coefficient | **83.91** | 83.60 | 82.65 |

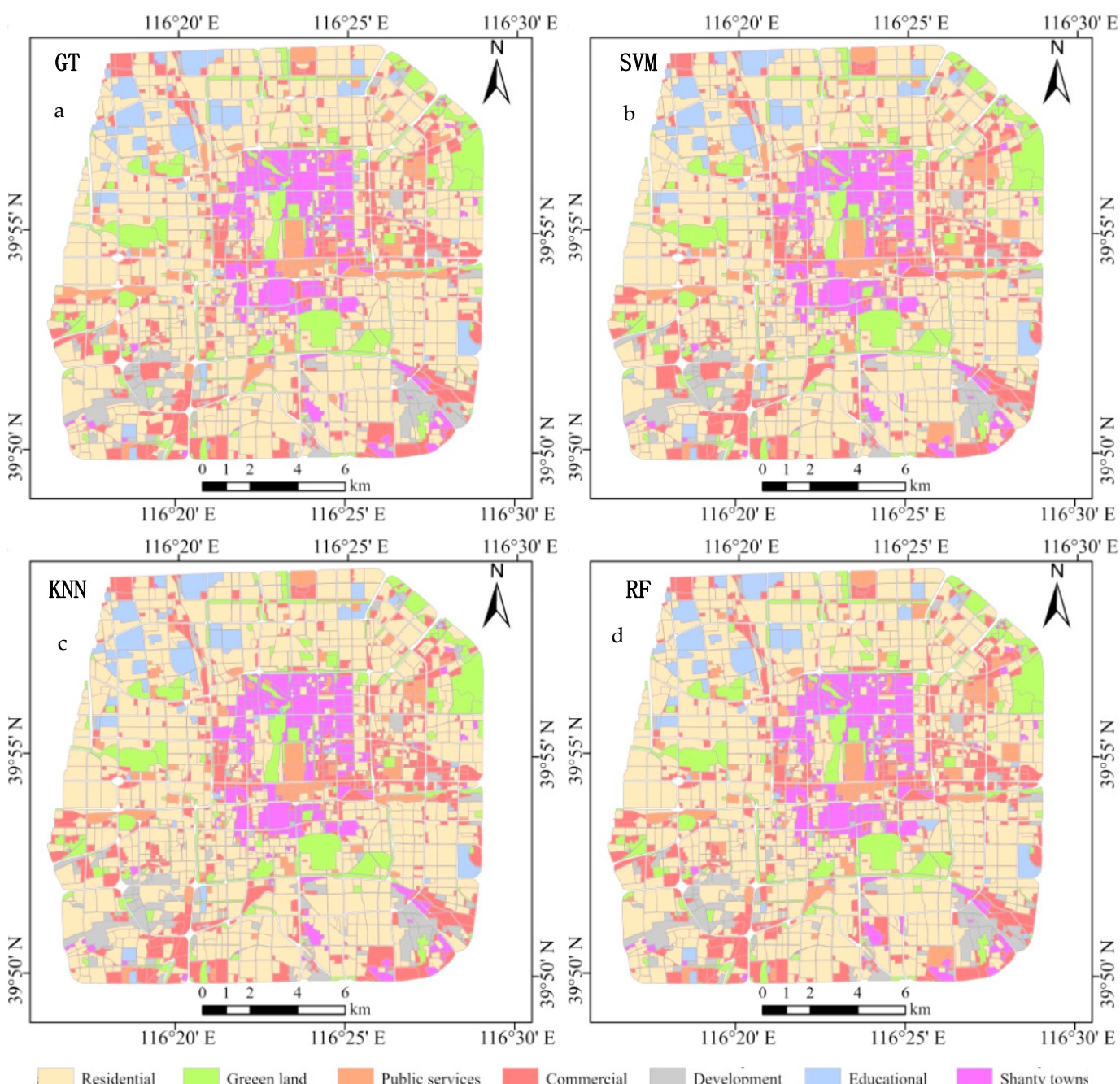

**Figure 9.** UFZs recognition results of different classifiers: (**a**) ground truth, (**b**) SVM recognition results, (**c**) KNN recognition results, and (**d**) RF recognition results.

### 4.2. Discussion

#### 4.2.1. Comparison of Different Pre-Training Models

To compare the performance of different pre-trained models in identifying UFZs, ST-CNN selected AlexNet, VGG16, ResNet50, and MobileNet-V2 as pre-training models, respectively [27,63–65]. The classifier is softmax. In order to integrate with the self-built network, we delete the network structure behind the first fully connected layer of the pre-trained model and add a fully connected layer that outputs 64 dimensions. Furthermore, two training patterns were employed for the pre-trained model: training with all full parameter layer and training with only the full connection layer. A comparison was made between these two patterns. As shown in Figure 10, the training pattern that involved all parameter layers yielded superior results compared to training with only full connec-

tion layers. Specifically, when using pre-trained VGG16, the OA and kappa coefficient increased by 5.77% and 7.23%, respectively. In the model of all parameter layer training, the OA of the pre-trained VGG16 increased by 7.56%, 2.79%, and 5.37% compared with pre-trained AlexNet, ResNet50, and MobileNet-V2, respectively, and the kappa coefficient increased by 9.96%, 3.60%, and 6.85%, respectively. Therefore, ST-CNN can obtain the optimal classification results of UFZs by using the pre-trained VGG16 and all parameter layer training.

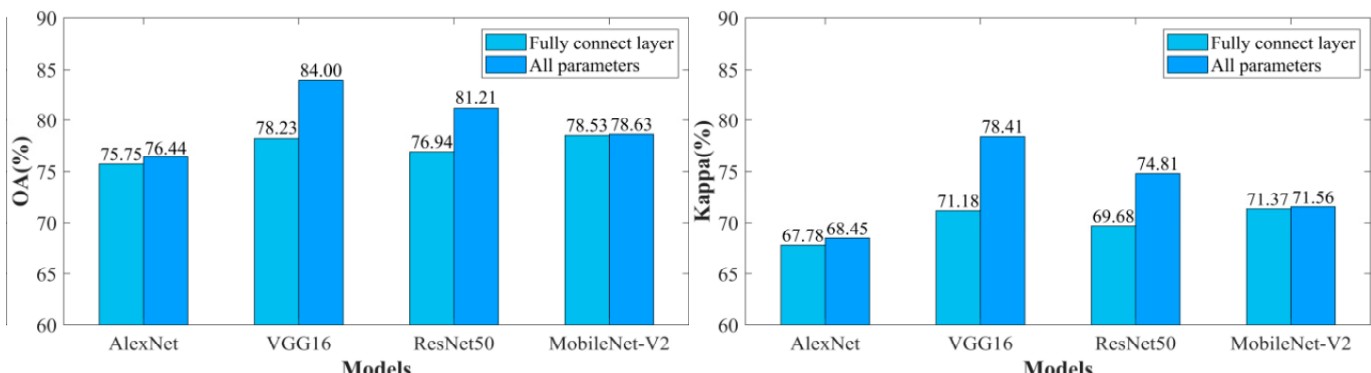

**Figure 10.** Results of CNNs with different training models.

Compared to AlexNet, VGG16 utilizes a series of continuous 3 × 3 Convs instead of 7 × 7 Convs. Due to the small parameter number of 3 × 3 Convs, VGG16 has a deeper and wider structure, which can extract more features and fit the network effectively. In addition, ResNet50 and MobileNet-V2 do not show obvious advantages. On the one hand, gradient dispersion may occur in deep networks, making it difficult to optimize the model. On the other hand, there are more parameters in the deeper network, and the number of images in the dataset in this paper is small, so the whole network cannot be trained effectively.

4.2.2. Advantages of Semi-Transfer Structure

To demonstrate the advantages of the semi-transfer structure, we design four models as shown in Table 5. The classifier used in all models is softmax. M1 represents a model whose input is RGB images and the network structure is the pre-trained VGG16. M2 indicates that the input is RGB + NIR + nDSM images and the network structure is the self-built CNN. M3 indicates that the input is RGB images, the network structure is ST-CNN, and the input of both pre-trained VGG16 and self-built CNN is RGB images. M4 represents a model whose input is RGB + NIR + nDSM images and the network structure is ST-CNN, the input of pre-trained VGG16 is RGB images and the input of self-built CNN is RGB + NIR + nDSM images. From Table 4, M3 and M4 with ST-CNN network structure have higher OA and kappa coefficients than M1 and M2. M4 obtained the highest OA and kappa coefficient. Figure 10 shows the training curves of different models. From Figure 11, M4 can effectively combine the self-built CNN with the pre-trained VGG16 network. Speed up model training while avoiding overfitting caused by small amounts of data. Compared with M1, the OA and kappa coefficient of M3 increased by 0.3% and 0.38%, respectively. This is due to ST-CNN using parallel network designs of different depths, which can extract different depths of image features and improve recognition accuracy.

**Table 5.** Accuracy of different models for identifying UFZs.

| Model | Network | Input | OA (%) | Kappa Coefficient |
|---|---|---|---|---|
| M1 | Pre-trained VGG16 | RGB | 81.71 | 75.56 |
| M2 | Self-built CNN | RGB + NIR + nDSM | 75.84 | 67.92 |
| M3 | ST-CNN | RGB | 82.01 | 75.94 |
| M4 | ST-CNN | RGB + NIR + nDSM | 84.00 | 78.41 |

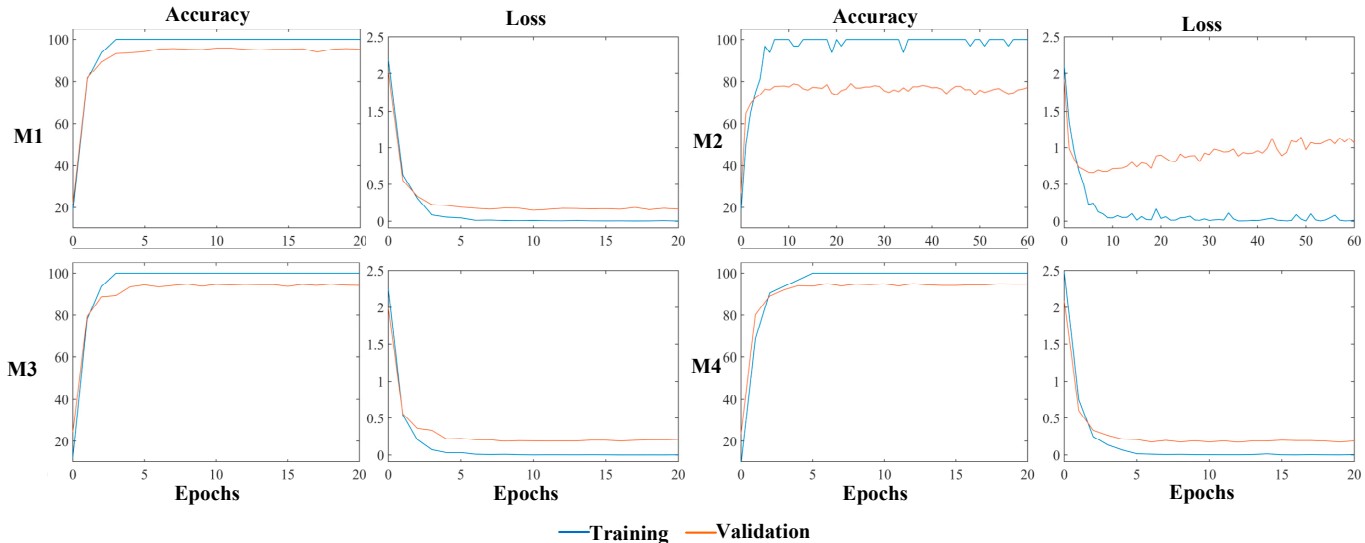

**Figure 11.** Training curves of different models. M1: pre-trained VGG16 (RGB); M2: self-built network (RGB + NIR + nDSM); M3: ST-CNN (RGB); M4: ST-CNN (RGB + NIR + nDSM).

### 4.2.3. Contribution of Different Modality Data

We propose a multimodal data fusion framework for UFZs identification. To analyze the impact of RGB, NIR, nDSM, and POI data on identifying UFZs, we designed five different inputs. The network structure is ST-CNN, and the classifier is SVM. Table 6 presents the recognition results of UFZs with different inputs.

**Table 6.** Comparison of UFZs recognition results with different inputs (N: NIR, D: nDSM, P: POIs).

|                    | RGB   | RGB + N | RGB + D | RGB + N + D | RGB + N + D + P |
|--------------------|-------|---------|---------|-------------|-----------------|
| Residential zones  | 86.81 | 89.69   | 89.93   | 90.41       | 94.00           |
| Commercial zones   | 78.66 | 79.50   | 80.75   | 79.92       | 83.68           |
| Shantytown         | 90.20 | 88.24   | 91.18   | 92.16       | 92.16           |
| Public service     | 40.54 | 31.08   | 29.73   | 37.84       | 55.41           |
| Development        | 91.07 | 94.64   | 91.07   | 94.64       | 96.43           |
| Education          | 35.29 | 41.18   | 47.06   | 47.06       | 64.71           |
| Green land         | 95.05 | 95.05   | 92.08   | 93.07       | 94.06           |
| OA                 | 82.01 | 82.80   | 83.00   | 84.00       | 88.17           |
| Kappa coefficient  | 75.94 | 76.82   | 77.03   | 78.41       | 83.91           |

Table 6 shows the effectiveness of incorporating multimodal data in improving the recognition accuracy of UFZs. When comparing the RGB to the additional NIR data, there is an increase of 0.79% in OA and 0.88% in kappa coefficient. The recognition accuracy of residential zones, commercial zones, development, and education has improved, with enhancements ranging from 0.84% to 5.89%. The inclusion of the nDSM data leads to 0.99% in OA and 1.09% in kappa coefficient. The recognition accuracy of residential zones, commercial zones, education, and shantytown increased, ranging from 0.98% to 11.77%. When adding NIR + nDSM data, the OA and kappa coefficient increased by 1.99% and 2.47%, respectively. The recognition accuracy increases for residential zones, commercial zones, development, education, and shantytown ranging from 1.26% to 11.77%. Compared with the input of RGB + NIR + nDSM, when POIs are added, the OA and kappa coefficient have been greatly improved, increasing by 4.17% and 5.50%, respectively. With the exception of shantytowns, the recognition accuracy for other UFZ categories experiences improvements ranging from 0.99% to 17.65%.

After adding NIR and nDSM data, the OA and kappa coefficient of UFZs improve. However, the recognition accuracy varies across different categories of UFZs, which may

be attributed to the different urban landscapes associated with each category. For instance, nDSM data enhance the recognition accuracy of residential zones, commercial zones, education, and shantytown with dense buildings. POI data showcase high-level semantic information tightly related to human economic activities, which can improve the recognition accuracy of UFZs, especially for public services and education. Therefore, social perception data such as POIs are important for UFZs identification. However, it should be noted that even with the inclusion of nDSM and POIs, the accuracy of identifying public services and education remains relatively low. For public services, the diversity of POI types associated with public services reduces the effectiveness of the extracted POI semantic characteristics. Additionally, some buildings within the public service are similar to commercial buildings in the image, resulting in similar image features. For education, the low accuracy can be attributed to the small number of original training samples, which may limit the model's generalization ability. Furthermore, educational buildings often exhibit similarities with residential buildings, such as dormitory buildings.

### 4.2.4. Impact of PCA Dimensionality Reduction on Identifying UFZs

We utilize the PCA algorithm to reduce the dimensionality of the 128-dimensional image features extracted by ST-CNN and combine them with POI semantic features. To determine the optimal dimension after PCA dimension reduction, we conducted a comparative analysis by reducing the image features to 1–127 dimensions and compared them with the original 128-dimensional features, as shown in Figure 12. From Figure 12, the cumulative variance contribution rate of the top five image features has exceeded 80%. As the image dimension increases, the OA and kappa coefficient of UFZs identification initially show a rapid increase followed by a slower decrease. Notably, when reducing the image features to 15 dimensions, UFZs recognition achieves the highest OA of 88.17%, and the kappa coefficient is 83.91%. When the image features are reduced to eight dimensions, the kappa coefficient of UFZs recognition is the highest, reaching 83.96%, and the OA is 88.07%. Considering that the difference in OA is greater than the difference in kappa coefficient, we choose 15 dimensions as the best dimension for dimensionality reduction.

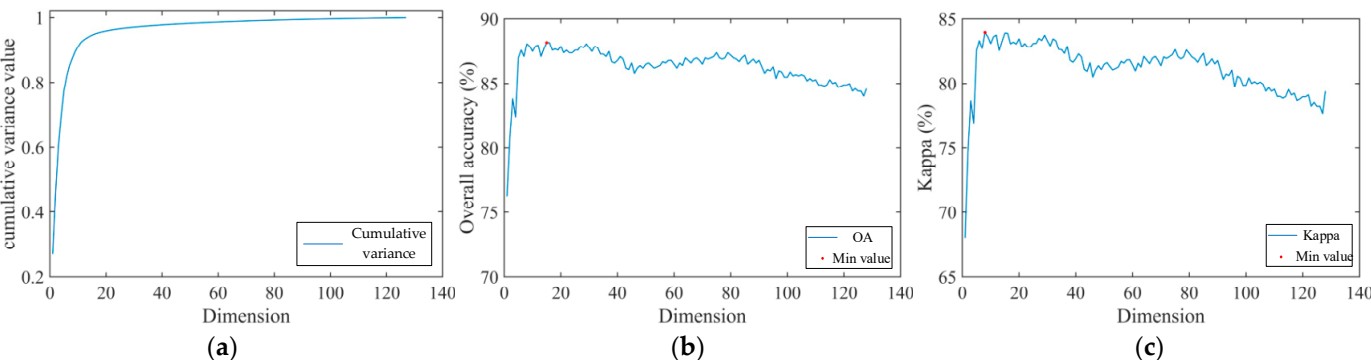

**Figure 12.** Cumulative variance contribution rate and recognition accuracy of image features in different dimensions: (**a**) cumulative variance contribution rate, (**b**) overall accuracy, and (**c**) kappa coefficient.

### 4.2.5. Compare with Other Methods

To further validate the feasibility of the proposed method, a comparison was made with previous studies on UFZs identification in Beijing, as shown in Table 7. Most of these studies fuse remote sensing images and data that can reflect human activities to identify UFZs. In cities with complex functions, such fusion approaches generally exhibit higher accuracy compared to using remote sensing images. However, it is worth noting that while medium-resolution remote sensing images are more readily available for large-scale UFZs identification, their recognition accuracy tends to be lower than that achieved with high-resolution images due to the lack of detailed ground object information. In

addition, our method demonstrates slightly lower accuracy than the second and fourth methods. This is primarily due to the lower identification accuracy in the categories of public service and education, which limits the OA of UFZs identification. However, when compared to the first and the third methods, our method achieves 4.2% and 7.2% higher OA, respectively. Overall, although our method does not attain optimal classification accuracy, it still possesses certain advantages, particularly in terms of its simplicity and ease of model training.

**Table 7.** Comparisons of the existing methods for UFZs identification.

| Method | Data Source | Study Area | Spatial | OA |
|---|---|---|---|---|
| Integrating bottom-up classification and top-down feedback [66] | WorldView-II image | Beijing, China (67.1 km$^2$) | Residential, commercial, shantytown, industrial, campuses, park | 84% |
| Hierarchical semantic cognition [67] | QuickBird image; POIs | Beijing, China (67.1 km$^2$) | Residential, commercial, shantytown, industrial, campuses, park | 90.8% |
| Similarity measures and threshold [68] | Landsat8 image; POIs | Beijing, China (16,808 km$^2$) | Level I classes: agriculture, green space, waterbody, undeveloped, residential, commercial, industrial, institutional | 81.0% |
| Integrating high spatial resolution nighttime light and daytime multi-view imagery based on B-OVW model [69] | Ziyuan3 (ZY3-01) im-age Jilin1-07 (JL1-07) im-age | Beijing, China (300 km$^2$) | Residential, commercial, shantytown, industrial, campuses, park, and green space | 89.6% |
| Our method | GF-7 image; POIs | Beijing, China (300 km$^2$) | Residential, commercial, shantytown, public service, development, education, green land | 88.2% |

## 5. Conclusions

In this paper, we proposed a novel "Image + Text" framework for UFZs recognition by integrating GF-7 multi-spectral images, urban 3D information, and POI social perception data. The framework utilizes ST-CNN for extracting image features and LDA for extracting semantic features. ST-CNN benefits from a pre-trained model and requires only a small number of samples, while effectively incorporating the multi-spectral and multi-dimensional features of NIR and nDSM images. The experimental results showed that the proposed framework enhances the accuracy of UFZs identification and accelerates the model training process. The inclusion of NIR, nDSM, and POI data can improve the identification accuracy of UFZs. In the future, we will explore automatic segmentation methods of urban functional units [70], introduce social perception data such as nighttime light images and street view images [39,71], and explore more effective multimodal data fusion strategies.

**Author Contributions:** Conceptualization, Z.S.; methodology, P.L. and Z.S.; validation, Y.S. and P.L.; writing—original draft preparation, Z.S. and P.L.; writing—review and editing, D.W., Q.M., and W.Z.; funding acquisition, Z.S. and D.W. All authors have read and agreed to the published version of the manuscript.

**Funding:** This research was funded by the Tianjin Municipal Education Commission Scientific Research Program (grant number 2021SK003); the Tianjin Educational Science Planning Project (grant number EHE210290); the Tianjin outstanding science and Technology Commissioner Project (grant number 22YDTPJC00230); and the National Natural Science Foundation of China (grant number 41971310). National Natural Science Foundation of China (grant number 42171357); National Key Research and Development Program of China (grant number YFC3301602).

**Institutional Review Board Statement:** Not applicable.

**Informed Consent Statement:** Not applicable.

**Data Availability Statement:** Not applicable.

**Acknowledgments:** We appreciate the constructive comments and suggestions from the reviewers that helped improve the quality of this manuscript.

**Conflicts of Interest:** The authors declare no conflict of interest.

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
