# Peer review of "Recognizing Urban Functional Zones by GF-7 Satellite Stereo Imagery and POI Data"

_applsci, doi:10.3390/app13106300_

Round 1
Reviewer 1 Report
Dear authors,
thank you for the opportunity to review this work. The manuscript presents an innovative method to recognize the urban functional zones by using GF-7 Satellite Stereo Imagery and POI Data, certainly in accordance with the scope of Applied Sciences. The manuscript includes an introduction with a description of the importance of an accurate and rapid identification of UFZs and the existent studies, the description of the study area and the methodology used, the discussion on the obtained observations and results and the conclusions.
Minor comments below:
I suggest that authors to check that all acronyms have the same style, making sure to put the first letter in block letters (e.g., urban functional zones (UFZs), red-green-blue (RGB), normalized digital surface model (nDSM), probabilistic topic mode (PTM), scale-invariant feature transform (SIFT), etc.)
For better visualization, enlarge the font of the titles and axes of some figures (e.g., Figure 10, the axes of Figure 11, Figure 12)
The reading is fluent and easy to understand
Author Response
Dear reviewers, please refer to the attachment. The modified part of the opinion is marked in yellow.

Reviewer 2 Report
This manuscript explores the “Recognizing urban functional zones by GF-7 Satellite Stereo Imagery and POI data”. The manuscript is elaborately described and contextualized with the help of previous and present theoretical background. All the references cited are relevant to this area of research. The methods/analytical study are clearly stated. The result and discussion section are clearly presented. The manuscript needs the following modifications before the acceptance.
1. Abstract: Present your research recommendation.
1. Arrange the key words in alphabetical order
2. What is the novelty of your research?
3. Materials and Method section is too lengthy.
4. Fig.11,11,12 are not clear. Provide a clear image.
5. Compare your results with existing studies.
6. Combine section 4 and 5 and provide the title ‘Result and Discussion’
7. Conclusion: Mention the research recommendations and scope for the future work
8. Remove the heading 7. Patent.
Minor editing of English language required
Author Response

(The authors gave the same response as above.)

Reviewer 3 Report
Dear authors!
I can see that you have made a great job and a lot of effort with your work and this paper itself.
The subject and your research seems important and innovative.
The structure of the article is correct, you have presented all necessary steps while describing the topic. All definitions were included.
From my point of view, there are only minor things that need to be improved:
1. Line 19-20, “The semantic features of Point of Interest (POI) obtained 19 based on probabilistic methods ignore the underlying semantic information” – there is something wrong with this sentence
2. Line 124 - it would be better to use phrase: "we have chosen" or "the chosen area"
3. Line 166 - it would be good to explain – why you have chosen the second level for classification standard.
4. Line 209 – don’t start the sentence with “And”
5. Necessity to provide list of abbreviations, as not all of them are explained (i.e. Relu layer, BN layer etc.)
The language seems fine, apart from the suggestions that were underlined in the revision form. There are only minor mistakes.
Author Response

(The authors gave the same response as above.)

Reviewer 4 Report
I made some notes mainly regarding the illustrations that, in some cases should be improved. The same, I would suggest for the English that is sometimes not so easy to read. I would also suggest to improve the conclusion chapter and add references already reported in the text
Some notes about the illustrations
Figure 1. Should be framed into the wider picture of entire China. Where is this place located?
Figure 4. Some of the text inside the illustrations is too small to read
Figure 8. I think the the caption should be improved to explain more accurately the meaning of the diagrams The conclusion should be improved and some references added
Please check the entire text
Author Response

(The authors gave the same response as above.)

Reviewer 5 Report
In this paper, the authors provided sound justification for their methodology to utilize both the GF-7 multi-view multi-spectral images and POI data to effectively and efficiently identify UFZs. Overall, I found this paper interesting and informative but I have a few comments to share with the authors:
Line 147-149: spell out the acronyms (GCP, TP, SGM) the first time they are introduced.
Figure 3(b): I’m not quite sure how to read the POI Density values in the map legend. What is the area unit (as in number of POIs per ???) and what is the resolution of the raster grid cell?
In section 3.1 and Figure 5, how were the urban functions (e.g., commercial, residential) determined at this stage before the block boundary was corrected?
Line 219, what is the resolution of the pixel?
In section 5, discuss why certain urban functions (e.g., residential) are more accurately recognized than others (e.g., education) even though the overall recognition accuracy was improved using inputs from RGB+N+D+P.
Round 2
Reviewer 4 Report
Thank you for all the corrections and the improved illustrations